# The Potential of *Parapanteles hyposidrae* and *Protapanteles immunis* (Hymenoptera: Braconidae) as Biocontrol Agents for the Tea Grey Geometrid *Ectropis grisescens* (Lepidoptera)

**DOI:** 10.3390/insects13100937

**Published:** 2022-10-16

**Authors:** Zi-Qi Wang, Xiao-Gui Zhou, Qiang Xiao, Pu Tang, Xue-Xin Chen

**Affiliations:** 1Institute of Insect Sciences, College of Agriculture and Biotechnology, Zhejiang University, Hangzhou 310058, China; 2Zhejiang Provincial Key Laboratory of Biology of Crop Pathogens and Insects, Zhejiang University, Hangzhou 310058, China; 3Ministry of Agriculture Key Lab of Molecular Biology of Crop Pathogens and Insects, Zhejiang University, Hangzhou 310058, China; 4State Key Lab of Rice Biology, Zhejiang University, Hangzhou 310058, China; 5Ministry of Agriculture Key Laboratory of Tea Quality and Safety Control, Tea Research Institute of Chinese Academy of Agricultural Sciences, Hangzhou 310008, China

**Keywords:** predominant parasitoid, tea grey geometrid, biology control, mutual interference, functional response

## Abstract

**Simple Summary:**

The tea grey geometrid *Ectropis grisescens* is a significant insect pest of tea plants in China. *Parapanteles hyposidrae* and *Protapanteles immunis* are *Ectropis grisescens* larval parasitoids. Here, we studied the parasitism performance of these two parasitoid species on different host densities under different temperatures as well as the interference effect of parasitoid density. We found that both parasitoid species, *Pa. hyposidrae* and *Pr. immunis*, exhibited a type II functional response towards the tea grey geometrid *E. grisescens* at four tested temperatures. With increasing the density of *E. grisescens* larvae, the number of parasitized larvae increased until a maximum was reached. *Pr. immunis* performed better than *Pa. hyposidrae* under higher temperatures. The parasitism rate by a single female parasitoid decreased with increasing parasitoid density at different temperatures, resulting in a reduction of searching efficiency. The findings of this study showed that *Pr. immunis* could be a better effective biocontrol agent than *Pa. hyposidrae* against the tea grey geometrid.

**Abstract:**

The tea grey geometrid *Ectropis grisescens* has long been a significant insect pest of tea plants in China. Two parasitoids, *Parapanteles hyposidrae* and *Protapanteles immunis* (Hymenoptera: Braconidae: Microgastrinae), are the most important parasitoids in the larval stage of *E. grisescens*. Yet, the potential of these two parasitoids for controlling the tea grey geometrid is not known. Here, we studied the parasitism performance of these two parasitoid species on different host densities under different temperatures as well as the interference effect of parasitoid density. The results showed that both parasitoid species, *Pa. hyposidrae* and *Pr. immunis*, exhibited a Type II functional response towards the tea grey geometrid *E. grisescens* at four tested temperatures. With increasing the density of *E. grisescens* larvae, the number of parasitized larvae increased until a maximum was reached. The highest number of hosts parasitized by *Pa. hyposidrae* or *Pr. immunis* reached 14.5 or 14.75 hosts d^−1^ at 22 °C, respectively. The estimated values of instantaneous searching efficiency (a) and handling time (h) for *Pa. hyposidrae* or *Pr. immunis* were 1.420 or 3.621 and 0.04 or 0.053 at 22 °C, respectively. *Pr. immunis* performed better than *Pa. hyposidrae* under higher temperatures. The parasitism rate by a single female parasitoid decreased with increasing parasitoid density at different temperatures, resulting in a reduction of searching efficiency. The findings of this study showed that *Pr.*
*immunis* could be a better effective biocontrol agent than *Pa. hyposidrae* against the tea grey geometrid.

## 1. Introduction

The tea plant is an evergreen shrub whose leaves and leaf buds are commonly used to produce the teas we enjoy [1,2]. The tea grey geometrid, *Ectropis grisescens* Warren, 1894 (Lepidopotera: Geometridae), is one of the most destructive chewing pests widely found in tea plantations in China [3]. Their larvae cause innumerable damage to the leaves of tea plants, resulting in severe yield losses [4,5,6]. Chemical insecticides have been the major method for controlling this pest over the last few decades. Long-term overuse of insecticides, on the other hand, has caused environmental pollution and insecticide resistance [7,8,9]. Furthermore, excessive pesticide residue on tea leaves is a serious problem faced by the tea production industry [10]. In view of these concerns, biological control is thought to be a more appropriate and reliable mean for insect pest control.

Parasitoids are considered the most effective natural enemies used in biological control programs [10,11,12,13]. It is essential to evaluate and test the efficacy of potential biological control agents under laboratory conditions before their release in the field. The functional response is regarded as a valuable measure for evaluating a natural enemy’s potential success [14]. It reflects the relationship between the number of preys or hosts attacked by a predator or parasitoid and the population density of the preys or hosts. When a parasitoid meets various densities of hosts, the functional response offers a quantitative explanation of the parasitoids’ behavior. Type I linear, Type II rectangular hyperbola, and Type III sigmoidal are the three basic forms of functional response [14,15,16,17]. Parasitoids are frequently linked to the Type II response. Meanwhile, various functional responses of the same species may be influenced by factors such as prey or host type, life stage, and temperature [18,19,20,21]. At high or low temperatures, the longevity and life-time fecundity of parasitoids are decreased [22]. Temperature also affects the functional response of a parasitoid, especially in the range of temperatures where parasitoids search for their hosts [23].

*Parapanteles hyposidrae* and *Protapanteles immunis* (Hymenoptera: Braconidae: Microgastrinae) are the most important parasitoids in the larval stage of the tea geometrid *E*. *grisescens* in the tea-growing areas of China. They can be easily separated from each other: *Pa. hyposidrae* has the head and mesosoma black, the metasoma dark brown, the abdominal segments with white areas, and its cocoon smooth and mostly yellowish–green while *P**r. immunis* has its body black with yellow hind femur, and its cocoon covered with fluffy cotton-like filaments and white [24]. However, it was frequently reported that they had variable parasitism rates and percentage of populations in the different localities and among the different years [24]. Therefore, we hypothesize that the performance of these two parasitoids may be affected by many factors, such as temperature, host population, and density of parasitoids themselves. To test our hypothesis, we studied the parasitism performance of these two parasitoid species on different *E. grisescens* densities under different temperatures as well as the interference effect of parasitoid density, to determine the potential of *Pa. hyposidrae* and *Pr. immunis* as biocontrol agents of *E. grisescens*. Our research results will provide basic data for biological control of *E. grisescens*.

## 2. Materials and Methods

### 2.1. Insect Cultures

*Ectropis grisescens*, *Parapanteles hyposidrae,* and *Protapanteles immunis* were collected from Yuhang district, located in Zhejiang Province, China (longitude 119.66667 E, latitude 30.15 N). All insect colonies were maintained in an environment-controlled room (21 ± 1 °C, 60 ± 10% RH and 16:8 h L: D photoperiod, designed by Zhejiang University, Model: AGC-1). The tea plant was obtained from the Tea Research Institute, Chinese Academy of Agricultural Sciences (longitude 120.09377 E, latitude 30.18 N) to rear *E. grisescens*. Second instar larvae of *E. grisescens* were used as hosts for the subculture of parasitoids. The adults of *Pa. hyposidrae* and *Pr.*
*immunis* were maintained in the same environment-controlled room and fed with 10–20% honey water at the same time.

### 2.2. Functional Response at Different Temperatures

Each experimental unit consisted of a plastic box (upper diameter 138 mm, lower diameter 107 mm, height 113 mm) with a circular hole that was a diameter of 6 mm, and the small hole was plugged with absorbent cotton. We tested the functional response of *Pa. hyposidrae* and *Pr. immunis* at four temperatures (18 °C, 22 °C, 26 °C, and 30 ± 0.5 °C) under 60 ± 5% relative humidity and 16:8 h light: dark photoperiod in plant growth chambers (Panosonic MLR-352H-PC). Different numbers of 10, 15, 20, 25, and 30 s instar larvae of *E. grisescens* were offered to one mated female (24 h old) of either *Pa. hyposidrae* and *Pr.*
*immunis* at each of four temperatures.

An appropriate number of fresh tea leaves was placed in the experimental containers. The duration of the experiment was 24 h and then the parasitoids were removed and the containers containing larvae were kept in incubators at the same constant temperatures. Under the same conditions, we continued to feed *E. grisescens* larvae, replaced fresh tea leaves every day, observed, and recorded the number of wasp cocoons and eclosions emerging until the host died from parasitization or pupates. Unhatched pupae were dissected under a stereomicroscope to examine whether they were parasitized. Each treatment with different combination of temperatures and densities was replicated four times.

### 2.3. Parasitoid Performance under Dynamic Parasitoid Density

The experimental environment and apparatus are the same as those in Section 2.2. 60 s instar larvae of *E. grisescens* were offered to various mated female (24 h old, 1, 2, 3, 4, and 5 adults) of either *Pa. hyposidrae* or *Pr. immunis* at two temperatures (22 °C and 26 °C). An appropriate number of fresh tea leaves were placed in the experimental containers. Each treatment with different combination of temperature and density was replicated four times.

### 2.4. Statistical Analyses

The functional–response data was analyzed with the R (version 4.1.2) package ‘Frair’ [25], which employed a functional response model based on logistic regressions. Logistic regression analyses of the proportion of parasitized larvae as a function of initial density were used to determine the type of functional response [26]. The coefficient of linear term (*p* = 0, a linear increase in parasitism rate as host densities rise, until reaching a maximum parasitism rate) characterized Type I, a negative first-order term (*p* < 0, declining proportional consumption with increasing resource density) characterized Type II, while a positive first order term followed by a negative second-order term characterized Type III (*p* > 0, initial increase and subsequent decrease in proportional consumption) [26]. The data were fitted to the Type I model described by Holling [25]:(1)$$N_{e}=aN_{0}T$$

The ‘frair_test’ function was used to assess whether a Type II or III functional response better reflected the relationship between host density and the number of parasitized hosts or each trial combination.

Significant negative linear coefficients from logistic regression suggest Type II functional response model described by Rogers [27]:(2)$$N_{e}=N_{0}\left[1-\sqrt{a\left(N_{e}h-T\right)}\right]$$

Significant positive linear coefficients from logistic regression suggest Type III functional response model described by Real [28]:(3)$$N_{e}=N_{0}\left[1-\sqrt{bN_{0}^{q}\left(N_{e}h-T\right)}\right]$$

A Type II response is defined by the parameters *a* (the instantaneous searching efficiency or attack rate), *h* (the handling time required for parasitic wasps to parasitize or attack a host), and *T* (experimental duration, 24 h), whereas a Type III response is defined by the parameters *h*, *q* (scaling component), *b* (search coefficient), and *T*. The scaling component *q* is a critical determinant of the functional response shape. It shows the extent to which functional response changes from a decelerating hyperbola to a sigmoidal form, with *q* = 0 in Type II and *q* > 0 in Type III. Following these analyses and since our data fit a Type II functional response, we used the Rogers Type II equation. The component parameters (*a* and *h*) were compared by using the “difference method” outlined in Juliano [24] via “frair compare” z-tests and of optimized coefficients, adopting the significance level of 5% (*p* < 0.05) for all statistics.

Bootstrapping was used to construct 95% confidence intervals to visualize variability around the fitted curves. Based on the output from bootstrapped fits, a functional response curve could be constructed using the ‘drawpoly’ function of the ‘Frair’ package.

To detect mutual interference among the parasitoid wasp, data were analyzed using a one-way ANOVA, and multiple comparisons of means were carried out with Tukey’s HSD test.

The equation by Hassell and Varley [29]:(4)$$\log_{10}a=\log_{10}Q-m\log_{10}P$$

We call the slope *m* the mutual interference constant, the parameter *Q* is the quest constant, and log_10_*Q* is the intercept of the equation. The parameter *a* is the search rate of the parasitoid wasp and was estimated for each replicate as:(5)$$a=\left(\frac{A}{P}*T\right)\ln\left(\frac{N_{0}}{N_{S}}\right)$$

The parameter *P* is the number of parasitoids, *T* is the duration of the experiment (24 h, also often taken to be (1), *A* is the total area (often assumed to be 1), *N*_0_ is the number of hosts (60), and *N_s_* is the number of hosts not parasitized.

## 3. Results

### 3.1. Parasitoid Performance under Dynamic Host and Different Temperatures

The results of the factorial analysis showed that temperature and host density had a significant effect on the proportion parasitized. The number of the *E. grisescens* larvae parasitized by *Pa. hyposidrae* or *Pr.*
*immunis* increased with the increase of temperatures and densities of the host *E. grisescens* larvae. According to the logistic regression analysis, females of parasitoids exhibited a Type II functional response when attacking second instar larvae of *E. grisescens* at various temperatures, because the parasitized *E. grisescens* larvae against the initial density of host larvae yielded significantly negative linear parameters (*p* < 0) for *Pa. hyposidrae* (*Pr*(*z*) < 0.05; Table 1) and *Pr. immunis* (*Pr*(*z*) < 0.001; Table 1). The functional response curves (Figure 1a,b) indicated that the number of *E. grisescens* larvae parasitized increased with an increase of the number of host individuals offered until a maximum was reached at all the temperatures tested. Therefore, we concluded that both parasitoid species exhibited Type II functional responses when parasitizing *E. grisescens* larvae.

In the *Pa. hyposidrae*–*E. grisescens* parasitoid–host combination (Figure 1a), as the temperature increased from 18 °C to 30 °C, the maximum number of hosts parasitized increased from 2.5 hosts d^−1^ to 14.5 hosts d^−1^, then decreased to 11 hosts d^−1^, and finally decreased to 4.75 hosts d^−1^. The results indicated that searching efficiency changed from 0.613 h^–1^ at 18 °C and 0.653 h^–1^ at 30 °C to 1.535 h^–1^ at 26 °C and 1.420 h^–1^ at 22 °C. The handling times were calculated to be 0.153 h, 0.040 h, 0.063, and 0.173 h, at 18, 22, 26, and 30 °C, respectively (Table 1). However, in the *Pr.*
*immunis*–*E. grisescens* parasitoid–host combination (Figure 1b), the highest number of parasitized hosts was 14.75 hosts d^−1^ at 22 °C, followed by 26 °C and 30 °C, and the lowest number was at 18 °C. The results indicated that searching efficiency changed from 1.822 h^–1^ at 18 °C and 2.278 h^–1^ at 26 °C to 3.621 h^–1^ at 22 °C and 2.469 h^–1^ at 30 °C. The handling times were calculated to be 0.087 h, 0.053 h, 0.071, and 0.055 h, at 18, 22, 26, and 30 °C, respectively (Table 1).

The results for the comparison of Type II functional response parameters showed that *Pr. immunis* exhibited a higher number of parasitized hosts and instantaneous searching efficiency *a* than *Pa. hyposidrae* at all temperatures (Figure 1c–f). However, differences were not significant except at 22 °C by using the “difference method” (Table 2). Unlike the “difference method”, bootstrapping results show significant differences at 22 °C and 30 °C (95% CIs clearly overlap) (Table 3). At 30 °C, the handling time *h* of *Pa. hyposidrae* and *Pr. immunis* differed significantly (95% CIs no overlap) (Table 3).

### 3.2. Mutual Interference at Different Temperatures

We observed that parasitism rate by single female parasitoid decreased with increasing parasitoid density. The resulting mutual interference constant (*m*) for *Pa. hyposidrae* and *Pr. immunis* at 22 °C was estimated to be 0.304 or 0.245, respectively (Table 4), while at 26 °C the constant was estimated to be 0.210 or 0.260. Mutual interference among parasitoids reduces the searching efficiency with increasing parasitoid population density (Figure 2).

## 4. Discussion

The parasitoid functional response is regarded as crucial to host–parasitoid dynamics [30]. Our study is the first research to define the parasitoid functional response types of *Pa. hyposidrae* and *Pr. immunis* on *E. grisescens*. The significantly negative linear parameters (*p* < 0) obtained in the present study confirm that both parasitoids display Type II functional responses towards *E. grisescens*. Interestingly, *Pr. immunis* parasitized a higher number of hosts and exhibited higher instantaneous searching efficiency than *Pa. hyposidrae* at all temperatures. The handling time estimated for *P**r. immunis* was generally shorter than those for *Pa. hyposidrae* at higher temperatures, especially at 30 °C, which suggests that *Pr. immunis* could parasitize more *E. grisescens* at higher temperatures than *Pa. hyposidrae*. Therefore, *Pr. immunis* may be more effective for biological control of *E. grisescens*. The results are consistent with the data observed in the field that *Pr. immunis* was the predominant parasitoid of *E. grisescens* by investigating the species and number of parasitoids of *E. grisescens* (data unpublished).

In the functional response of Type II, the proportion of host consumed declines as host density increases [15]. The Type II functional response is theoretically less capable of suppressing host density when compared to the Type III functional response. Although Type II functional responses are common in host–parasitoid systems [30,31,32,33,34,35,36,37] and Type III is not prevalent in parasitic insects [34,38,39,40,41], the form of the functional response on its own does not determine the success or failure of parasitoids in biological control [30]. Other factors that influence the efficiency of natural enemies in pest control include prey growth rates, behaviors, and distribution [42,43,44], as well as temperature [16] and host plant [45].

Mutual interference amongst parasitoids can lead to a reduced rate of parasitism of host populations as parasitoid density increases [46,47]. We investigated the mutual interference of *Pa. hyposidrae* and *Pr. immunis* by observing and analyzing the parasitic behavior of single and multiple parasitoids, respectively. The results showed that the increasing number of parasitoids did not result in a proportional increase in the number of hosts parasitized, due to the negative effects of mutual interference. This is likely due to the limit space in the experimental arena that generated high conspecific encounter rates. As the density of conspecifics increases, each individual parasitoid spends less time searching for a host and more time interacting with other conspecifics [48], and this explains why one single female parasitoid parasitizes less hosts with the increase of parasitoid density. By fitting the data to the Hassell and Varley (1969) equation, the mutual interference constant (*m*) for *Pr. immunis* has no difference at 22 °C and 26 °C, while for *Pa. hyposidrae* the mutual interference constant estimated at 22 °C is higher than at 26 °C. Handling time or egg limitation of parasitoids influence the parasitism rate of parasitoids with a Type II functional response. The equation used to estimate the search rate does not take into account this influence, thus the mutual interference constant was consistently underestimated [49]. Mutual interference for a shared resource may have a consequence for the behavior of parasitoids, such as reducing the proportion of female offspring, which may change the population dynamics of the system, including higher host equilibrium densities and decreased stability [10,50,51].

In general, our study suggests *Pa. hyposidrae* and *Pr.*
*immunis* are active throughout the entire temperature range tested, therefore they may contribute markedly to *E. grisescens* suppression and could be incorporated into IPM systems that rely on natural enemies, as well as considered for augmentative releases.

## Figures and Tables

**Figure 1 insects-13-00937-f001:**
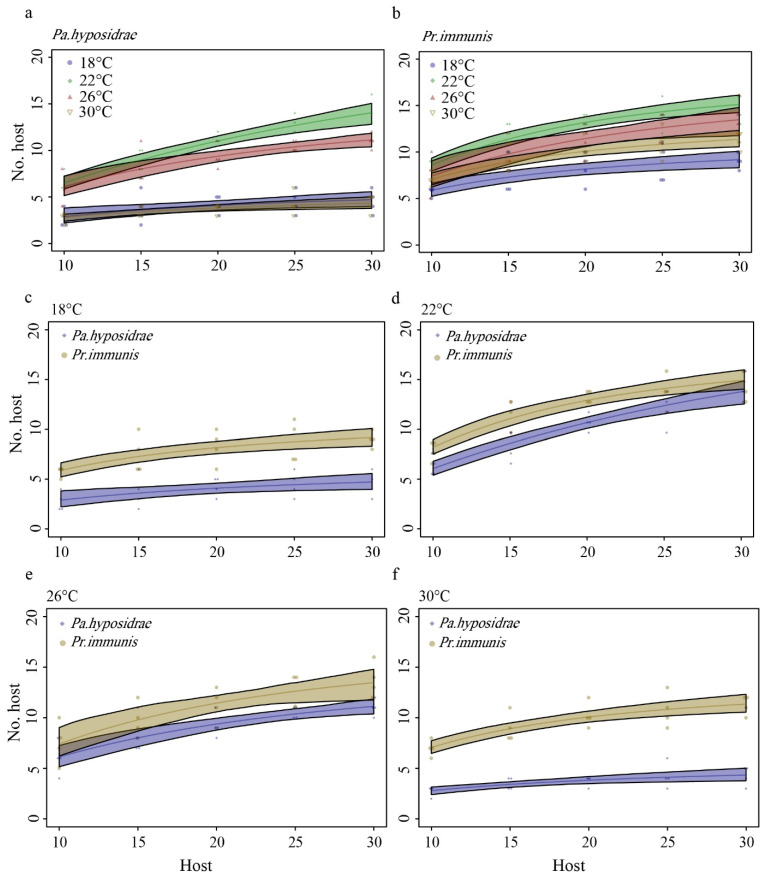
Functional responses of *Pa. hyposidrae* and *P. immunis* at five densities of *E. grisescens* under four temperatures. (**a**) *Pa. hyposidrae*, (**b**) *Pr. immunis*, (**c**) 18 °C, (**d**) 22 °C, (**e**) 26 °C, (**f**) 30 °C. Legends represent the observed values at four temperatures, x-axis represents the number of hosts, and y-axis represents the number of parasitized hosts. Curves show the predicted values based on Roger’s equation. Shaded areas are bootstrapped 95% confidence.

**Figure 2 insects-13-00937-f002:**
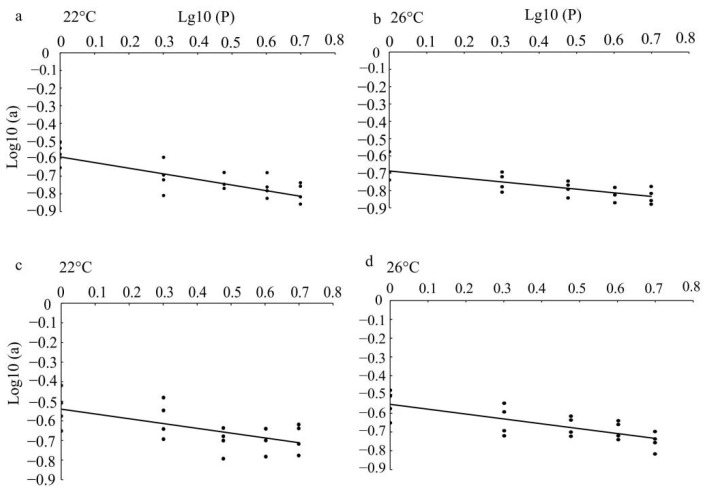
Regression lines of competition for *Pa. hyposidrae* and *P. immunis*. (**a**) and (**b**) represent interference relationship for *Pa. hyposidrae* at 22 °C and 26 °C, respectively. (**c**) and (**d**) represent interference relationship for *P. immunis* at 22 °C and 26 °C, respectively. Points represent the observed values. Lines show the predicted values.

**Table 1 insects-13-00937-t001:** Effects of temperatures on functional response parameters of *Pa. hyposidrae* and *Pr. immunis* towards *E. grisescens* based on the difference method model.

Wasp Species	T (°C)	Linear Coefficient		Type	Parameter	Estimate	SE	*z*-Value	*Pr*(*z*)
		*P*	*Pr*(*z*)						
** *Pa. hyposidrae* **	**18**	**−0.040**	**0.034**	**Type II**	** *a* **	**0.613**	0.422	1.451	0.146
					*h*	0.153	0.066	2.322	**0.020**
	22	−0.035	**0.0215**	Type II	*a*	1.420	0.420	3.383	**<0.001**
					*h*	0.040	0.014	2.817	**0.005**
	26	−0.048	**0.002**	Type II	*a*	1.535	0.562	2.730	**0.006**
					*h*	0.063	0.016	3.837	**<0.001**
	30	−0.041	**0.033**	Type II	*a*	0.635	0.499	1.271	0.204
					*h*	0.173	0.073	2.362	**0.018**
** *Pr. immunis* **	18	−0.057	**<0.001**	Type II	*a*	1.822	0.878	2.075	**0.038**
					*h*	0.087	0.019	4.580	**<0.001**
	22	−0.086	**<0.001**	Type II	*a*	3.621	1.027	3.526	**<0.001**
					*h*	0.053	0.007	7.571	**<0.001**
	26	−0.0578	**<0.001**	Type II	*a*	2.278	0.749	3.041	**0.002**
					*h*	0.055	0.011	4.959	**<0.001**
	30	−0.065	**<0.001**	Type II	*a*	2.469	0.985	2.508	**0.012**
					*h*	0.071	0.013	5.603	**<0.001**

Statistically significant differences in parameters (*Pr*(*z*) < 0.05) are indicated in bold. *P*, linear coefficient; *a*, the instantaneous searching efficiency; *h*, handling time; *Pr*(*z*), *p*-value; SE, standard error.

**Table 2 insects-13-00937-t002:** Estimates of the differences in parasitic rate (*D**a*) and handling time (*D**h*) between *Pa.*
*h**yposidrae* and *Pr. i**mmunis* under four temperatures.

Species Compared	T (°C)	Parameter	Estimate	SE	*z*-Value	*Pr*(*z*)
** *P* ** ** *a* ** **.** ** *h* ** ** *yposidrae* ** **vs.** ** *P* ** ** *r* ** **.** ** *i* ** ** *mmunis* **	18	*D* *a*	−1.208	0.974	−1.241	0.215
		*D* *h*	0.066	0.068	0.959	0.337
	22	*D* *a*	−2.201	1.108	−1.986	**0.047**
		*D* *h*	−0.014	0.016	−0.876	0.381
	26	*D* *a*	−0.742	0.937	−0.793	0.428
		*D* *h*	0.008	0.019	0.403	0.687
	30	*D* *a*	−1.835	1.104	−1.662	0.096
		*D* *h*	0.103	0.074	1.377	0.169

Significant pairwise differences (*p* < 0.05) are in bold.

**Table 3 insects-13-00937-t003:** Effects of temperature on functional response parameters of *Pa. hyposidrae* and *Pr. immunis* towards *E. grisescens* based on bootstrapped model.

Wasp Species	T (°C)	Parameter	CI Type	Lower	Upper
** *Pa. hyposidrae* **	18	*a*	BCA	0.296	2.761
		*h*	BCA	0.065	0.234
	22	*a*	BCA	0.990	2.116
		*h*	BCA	0.022	0.055
	26	*a*	BCA	0.886	3.046
		*h*	BCA	0.039	0.082
	30	*a*	BCA	0.378	1.280
		*h*	BCA	0.108	0.230
** *Pr. immunis* **	18	*a*	BCA	1.214	3.130
		*h*	BCA	0.069	0.104
	22	*a*	BCA	2.280	5.615
		*h*	BCA	0.043	0.061
	26	*a*	BCA	1.304	10.238
		*h*	BCA	0.035	0.081
	30	*a*	BCA	1.704	3.810
		*h*	BCA	0.058	0.083

*a*, the instantaneous searching efficiency; *h*, handling time.

**Table 4 insects-13-00937-t004:** Regression equation of mutual interference for *Pr. immunis* and *Pa. hyposidrae* at 22 °C and 26 °C.

Wasp Species	Parasitoids Density	Equation	Parameter	Estimate	SE	t	*p*
** *P. hyposidrae* **	22	log_10_a = −0.583 − 0.304log_10_P (R = 0.7927)	log_10_Q	−0.583	0.027	−21.8272	<0.001
			m	0.304	0.055	−5.516	<0.001
	26	log_10_a = −0.687 − 0.21log_10_P (R = 0.7454)	log_10_Q	−0.687	0.021	−32.055	<0.001
			m	0.210	0.044	−4.744	<0.001
** *P. immunis* **	22	log_10_a = −0.54 − 0.245log_10_P (R = 0.6286)	log_10_Q	−0.540	0.034	−15.668	<0.001
			m	0.245	0.071	−3.429	0.003
	26	log_10_a = −0.553 − 0.26log_10_P (R = 0.7514)	log_10_Q	−0.553	0.026	−21.257	<0.001
			m	0.260	0.054	−4.831	<0.001

SE, standard error.

## Data Availability

The data presented in this study are available on request from the corresponding author.

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
