# Peer review of "The Potential of *Parapanteles hyposidrae* and *Protapanteles immunis* (Hymenoptera: Braconidae) as Biocontrol Agents for the Tea Grey Geometrid *Ectropis grisescens* (Lepidoptera)"

_insects, 2022, doi:10.3390/insects13100937_

Round 1
Reviewer 1 Report
Thanks to the authors to report this interesting data. There are no major concerns with the paper. The methods used are appropriate for this type of study, the results are straightforward, the quality of the figures are very satisfactory, the reference list cover the relevant literature adequately, and the conclusions are supported by the data shown. Just the introduction section to be a bit expanded to better reflect the overall structure of the study. The written manuscript in its current form lacks a very clear and well-defined hypothesis and objectives at the end of the Introduction section. Please add a sentence or two providing insight and justifying why this subject is worth researching. Then, add the hypothesis that has been tested and the objective of the study. Furthermore, as a consequence of a clearly defined hypothesis and objective, the manuscript also lacks a conclusion at the end of the Discussion section. In the present form, the conclusion is generic and broad and will have to be rewritten. Without that information (hypothesis, objective, and conclusion), it is difficult to assess the payoff of this study and determine its contribution to the field!
Reviewer 2 Report
Dear Editor, authors,
The authors evaluated the functional responses of two parasitoids Pa. hyposidrae and Pr. Immunis at various temperatures and host densities, to control the tea grey geometrid, Ectropis grisescens in China. Although the paper is well-structured, it has many issues especially regarding the methodology.
1- Introduction needs to be rephrased particularly the second paragraph. In addition, more information is required regarding the two parasitoids.
2-My main concern is the methodology: as Pa. hyposidrae and Pr. Immunis were considered as Apanteles spp. so I would say that these two species are morphologically indistinguishable (visually)! Correct me if am saying wrong. In this case, it is recommended to rear the two parasitoids separately in two rooms. Do you have any information regarding the potential reproduction between the species. Are you sure that there was no contamination in the two cages? The parasitoids can easily get in the cages!!
Another main concern is the number of replicates. The authors conducted only 4 replicates for each experiment, which make the results statistically not correct. At least we need 7 or 8 replicates.
For the previous two issues, I cannot support the MS as a full paper. Otherwise, short communication might be good and so the authors must reduce significantly their MS.
In addition, English language could be improved especially in discussion.
Some specific minor corrections:
L25-26: based on your results, I would rather say only Pa. hyposidrae could be…..Since it is more adapted to temperatures than Pr. immunis
L58-70: The second paragraph is not well written and not connected to the previous or the following paragraph. You start directly by functional responses, an identification is needed here.
L58-59: Replace “Parasitoids are a type of pivotal natural enemies that play an extremely important role in pest biological control” by “Parasitoids are considered the most effective natural enemies that used in biological control programs”.
L62: Replace varied by various
L67: Not only high temperature, but the exposure to low temperature at immature stages may affect the longevity of adults.
L74 : E. grisescens: we start with the full name when starting a sentence.
L 74-80 : what is the objective of talking about the second species! If it is necessary, then you must move it to the first paragraph
L89-92: Put the rearing of the parasitoids after the rearing of E. grisescens.
L95: More information is needed regarding the parameters of functional Reponses of the two parasitoids.
-What kind of thermal incubators used in the experiments?
L101: delete “fully”. Otherwise, you can say one mated female
L102: delete “adult”.
L118: statistical analysis: the authors used two parasitoid species, four temperatures and five host densities. So I was wondering if an interaction could be considered in these models.
L130: the model is missing.
The authors started to use the term “Mutual interference from statistical anlaysis. Thus it is important to add some information in materials and methods.
L210-211: delete “The present study results show that mutual interference between parasitoids can be observed”.
L250: “environmental temperature”, but you study is on temperature.
L229: delete “to our knowledge”.
L229-231: rephrase
L233: “that are resistant to temperature influences”, what do you mean?
L255-257: yes, but you need to explain why!! Is there any intra competition among females of each parasitoid species. Is there any super-parasitism. Such information is important in this paragraph.
Figure 1: what does y-axis represent? I guess there is something wrong.
I suggest to read the following papers:
Tazerouni et al. 2019. Functional response of parasitoids: its impact on biological control
Ismail et al. 2014. Fitness consequences of low temperature storage of Aphidius ervi. Biocontrol
Hope the comments will be helpful
Reviewer 3 Report
This paper entitled “The potential of Parapanteles hyposidrae and Protapanteles immunis as biocontrol agents for the tea grey geometrid Ectropis grisescens” described a study of parasitism performance of two parasitoid species on different host densities under different temperatures. Using two logistic regression methods, the authors found that the parasitism performance changed dramatically in a temperature-dependent manner.
The tea plants are one of the most important economic crops in China, and the tea grey geometrid Ectropics grisecens is a significant insect pest of tea plants in major tea producing provinces in China. The authors showed that both parasitoids can be served as potential biocontrol agents which are effective to inhibit the survive of grey geometrid. The overall study is well designed, and the paper is clear-written with only a few spelling/grammar mistakes. I have the following comments/questions to improve the paper:
1. Explain why only type II and type III function responses were chosen to test parasitism performance not type I.
2. The formula of type II function response is missing in the method chapter.
3. A negative control is required to rule out the death of grey geometrids not due to the change of temperature, but for introducing parasitoids.
4. How the affection using parasitoids compared to using chemical insecticides?
5. It’s not clear to me how was the “handling time” defined in the method chapter.
6. More information is required for explaining Table 1: what’s the difference between Pr(z) under Linear coefficient and Pr(z) in the last column? What do “a” and “h” stand for?
7. Line229:delete “to our knowledge”
8. Move the formulas in line 152,156 in the center of the page.
9. Line110: change “density” to “densities”.
10. Line 270: change to ”temperature rises” .
11. Line 277: I don’t agree this study provides a “theoretical foundation” as this study does not reveal any mechanism underlying parasitoids that inhibit grey geometrid, but rather some evidence that they may serve as potential biocontrol reagents.
Round 2
Reviewer 2 Report
The authors have improved their paper. However, the big issue in this manuscript is the number of replicates. If I understood well, each replicate represented a female parasitoid of each species. So, four females for each treatment are not enough, and did not represent the whole population. Indeed, more replicates mean more variability and more reliability.
Thus, I will leave the decision to the editor.
Minor comments:
L65-68 “reference”. You may cite Tazerouni et al. 2019.
L64-65: Replace “It is essential to learn if a specific species of parasitoid wasp could be a potential biological control agent before it could be used in the field”.
By
“It is essential to evaluate and test the efficacy of potential biological control agents under laboratory conditions, before their release in the field”.
Author Response
Point 1: The authors have improved their paper. However, the big issue in this manuscript is the number of replicates. If I understood well, each replicate represented a female parasitoid of each species. So, four females for each treatment are not enough, and did not represent the whole population. Indeed, more replicates mean more variability and more reliability.
Thus, I will leave the decision to the editor.
Response 1: We agree with your opinion that more replicates mean more variability and more reliability. But we still thought four repeated experiments have been able to reflect the availability of the results in our study. The repeatability of our experimental results is very good. Three to four replicates are widely recognized in biological studies. For example, please see the papers as follows:
Wei Low, Robin W.J. Ngiam, Darren C.J. Yeo, Predation of mosquitos by odonates in a tropical urban environment: insights from functional response and field mesocosm experiments, Biological Control, 2021, 161, 104702, https://doi.org/10.1016/j.biocontrol.2021.104702. (3 replicates).
Ross N. Cuthbert, Jaimie T.A. Dick, Amanda Callaghan, James W.E. Dickey, Biological control agent selection under environmental change using functional responses, abundances and fecundities; the Relative Control Potential (RCP) metric, Biological Control, 2018, 121, 50-57. https://doi.org/10.1016/j.biocontrol.2018.02.008. (4 replicates).
Point 2: L65-68 “reference”. You may cite Tazerouni et al. 2019.
Response 2: added.
Point 3: L64-65: Replace “It is essential to learn if a specific species of parasitoid wasp could be a potential biological control agent before it could be used in the field”. By “It is essential to evaluate and test the efficacy of potential biological control agents under laboratory conditions, before their release in the field”.
Response 3: Replaced.

Reviewer 3 Report
I'm happy to see that all my questions have been addressed carefully by the authors, and I don't have any further questions prior to its publication.
Author Response
Thanks. We really appreciate your revision.